# Distinct Hormone Signalling-Modulation Activities Characterize Two Maize Endosperm-Specific Type-A Response Regulators

**DOI:** 10.3390/plants11151992

**Published:** 2022-07-30

**Authors:** Joaquín Royo, Luís M. Muñiz, Elisa Gómez, Ana M. Añazco-Guenkova, Gregorio Hueros

**Affiliations:** Departamento de Biomedicina y Biotecnología, Campus Universitario, Universidad de Alcalá, Alcalá de Henares, 28805 Madrid, Spain; joaquin.royo@uah.es (J.R.); lmmuniz2@hotmail.com (L.M.M.); elisa.gomezs@uah.es (E.G.); ana_macrina@hotmail.com (A.M.A.-G.)

**Keywords:** maize endosperm transfer cells, signal transduction, response regulator

## Abstract

ZmTCRR1 and 2 are type-A response regulators expressed in the maize endosperm transfer cells (TC). While type-B response regulators transcriptionally control canonical type-A response regulators, as part of the cytokinin signal transduction mechanism, the ZmTCRRs are regulated by ZmMRP1, a master regulator of TC identity. In addition, the corresponding proteins are not detected in the TC, accumulating in the inner endosperm cells instead. These features suggest these molecules are not involved in classical, cell-autonomous, cytokinin signalling pathways. Using transgenic Arabidopsis plants ectopically expressing these genes, we have shown that ZmTCRR1 and 2 can modulate auxin and cytokinin signalling, respectively. In Arabidopsis, the ectopic expression of ZmTCRR2 blocked, almost completely, cytokinin perception. Given the conservation of these signalling pathways at the molecular level, our results suggest that the ZmTCRRs modulate cytokinin and auxin perception in the inner endosperm cells.

## 1. Introduction

The cytokinin perception pathway best exemplifies canonical two-component signal transduction systems (TCS) in plants (reviewed in [1]). A plasma membrane receptor with histidine kinase activity auto-phosphorylates upon hormone binding. The signal moves through a chain of phosphate transfer reactions, first to a histidine phospho-transfer protein (HPT) and then to a type-B response regulator (type-B RR), a MYB-related transcription factor that regulates cytokinin-responsive genes. Type-A response regulators (type-A RR) are among the genes regulated by type-B RRs and act as negative modulators of the pathway, competing with the type-B RRs for phosphate transfer. Other signals can also regulate Type-A RRs allowing crosstalk between different regulatory pathways, including those initiated by members of the ethylene receptor family that possess functional histidine kinase activity and receiver domains [2].

ZmTCRR1 and ZmTCRR2 are maize genes with significant homology with type-A response regulators from bacteria, Arabidopsis, and rice. ZmTCRR1 expression presents a very narrow expression peak between 8 and 11 days after pollination (DAP), being hardly detectable at prior or later stages [3]. ZmTCRR2 presents a similar expression pattern, but it is still detectable up to 20 DAP [4]. Furthermore, ZmTCRR2 shows the canonical DDK triad in its peptide sequence, common to type-A RRs and is required to perform the phosphotransfer activity essential for TCS signal transduction [5]. Conversely, ZmTCRR1 presents a mutation that turns the triad’s second aspartic to histidine. This change is conserved in maize and its ancestor, Teosinte, and presumably interferes with the canonical phosphorylation mechanism [3].

Both genes appear to be controlled by ZmMRP1, a MYB-related transcription factor specifically expressed in the endosperm transfer cells (TC) during their earliest developmental stages [6,7]. Although the ZmMRP1 DNA binding domain is highly related to that present in type-B RRs, ZmMRP1 lacks any receiver motif suggesting its function as a TCS element. Thus, the module ZmMRP-1-ZmTCRRs seems to be an example of co-optation and adaptation of pre-existing machinery to new processes.

Both ZmTCRR genes are expressed in the transfer cell area at approximately 10 DAP, but the corresponding proteins accumulate in the developing endosperm storage area [3,4]. The role these genes might play in seed biology is currently unknown, albeit an involvement in signal transduction may be inferred from their similarity to other response regulators. This paucity of experimental evidence is partially due to the lack of mutants and the difficult accessibility of the endosperm transfer cells for experimentation.

The anatomical localisation of the ZmTCRR proteins and their expression timing suggested a possible role in tissue differentiation. All inner endosperm cells eventually develop into dead cells filled with storage products (starch and proteins), but the process is tightly regulated both temporarily and spatially [8]. The starch accumulation process precedes the accumulation of storage proteins and starts at 10–12 DAP, progressing from the upper part of the endosperm (the crown) towards the base. Programmed cell death (PCD) is first detected in the upper and central endosperm cells and affects the entire upper half of the endosperm by 28 DAP [9]. Remarkably, the inner basal endosperm cells positioned immediately above the transfer cells remain PCD negative as late as 36 DAP. This region, called conductive tissue [10], facilitates the symplastic transport of nutrients from the TC towards the upper endosperm. For this purpose, cells in the conductive tissue remain in a juvenile stage and, consequently, delay the processes of storage product accumulation and PCD. These facts led us to speculate that the ZmTCRR proteins might transmit signals from the mother plant into the inner juvenile endosperm cells [4].

The implication of TCS components in endosperm transfer cell differentiation and function has also been inferred in barley following transcriptomic analyses of laser-assisted microdissected transfer cells [11].

The function of TCS components in hormone signalling has been thoroughly studied in Arabidopsis, where many tools, transgenic lines, and mutants are available (reviewed in [12]. However, the position of the TC layer within the maize kernel complicates the study of this system in seed development. Furthermore, no mutants for ZmTCRR1 or 2 are available, and the generation of transgenic events to manipulate the system is cumbersome, requires a significant investment, and presents a relatively low success rate [13].

In this work, the ectopic expression of ZmTCRR1 and 2 in transgenic Arabidopsis revealed that these molecules have a divergent role in hormone signalling. While ZmTCRR2 is a strong negative modulator of cytokinin signalling, as expected for canonical type-A response regulators, ZmTCRR1 is a negative regulator of auxin. Assuming that in maize, the molecular function of these proteins resembles that observed in Arabidopsis, our data support a model in which the presence of both ZmTCRR proteins in the inner endosperm cells would influence their developmental pace by reducing their sensitivity to auxin and cytokinin signalling.

## 2. Results

### 2.1. Analyses of the Expression Patterns of ZmTCRR1 and 2

We previously determined the expression pattern of ZmTCRR-1 [3] and ZmTCRR-2 [4] using conventional techniques: Northern blot, RT-PCR, and RNA in situ hybridisation. However, nowadays, very detailed global expression atlases, based on RNAseq experiments, have become available for both endosperm development and endosperm domains. Therefore, we have extracted (Figure 1) the expression pattern of the ZmTCRRs from these datasets [14,15].

The RNAseq experiments essentially confirmed the expression pattern of the ZmTCRRs during kernel development that we previously obtained from RT-PCR analyses. Both genes are highly expressed in a very narrow timeframe between 6 and 10 DAP, with ZmTCRR2 being detectable, albeit at moderate levels, until 26 DAP.

However, the analysis of the distribution of the ZmTCRRs transcripts within the different kernel tissues (Figure 1B) produced unexpected results. While the data confirm transfer cell specificity for ZmTCRR1, the transcripts for ZmTCRR2 appeared at high levels in the TC and the conducting zone. This result potentially highlights a different functional role for these two molecules. Furthermore, since the transcription factor ZmMRP-1 is exclusively expressed in the TC, the expression of ZmTCRR2 in the conducting zone suggests divergent regulatory pathways for both ZmTCRRs.

### 2.2. ZmTCRRs Promoters Respond Differently to Hormonal Signals

Cytokinins transcriptionally regulate canonical type-A response regulators, the induction being detectable within minutes after treatment [16]. Therefore, we decided to test whether cytokinins and other hormonal inputs affected the ZmTCRR1 and 2 promoters in Arabidopsis, as an indication of their possible involvement in hormonal signalling in the maize seed.

As ZmTCRR1prom:GUS Arabidopsis plants have been described elsewhere [4], we focused on the characterisation of the ZmTCRR2 promoter activity in Arabidopsis. We cloned over 2000 bases of the 5′upstream region of ZmTCRR2 (located on chromosome 1 in maize; [4]) and fused it to a combined GFP:GUS construct. The reporter activity (Appendix A) was detected shortly after seed germination in an area close to the hypocotyl-root junction. From 7 days after germination, the expression extended to the rest of the hypocotyl and root, labelling the vessels intensively. At later stages, GUS activity localizes to leaf vessels and mesophyll cells. No signal was detected in flowers or siliques. Whole-mount observation under UV illumination showed the ZmTCRR2 promoter-driven GFP accumulated in the root stele at low levels and became more intense at the emergence points of lateral roots. No signal was visible at the root tip. We sectioned GUS-stained plants and localised the activity to the pericycle cells of the root vasculature. In summary, the ZmTCRR2 promoter is active in vessel-associated cells in the root of Arabidopsis, in contrast with the activity of the ZmTCRR1 promoter, which was restricted to the aerial parts of Arabidopsis, vessel-associated isolated cells, cotyledon tips, and leaf hydathodes [4].

We then examined the hormonal responses of both promoters. Transgenic plantlets were cultured for 7 days in MS and then 24 h in MS media supplemented with 1 µM ACC, BA, IAA, or GA3. Under these conditions (Figure 2), the ZmTCRR1 promoter showed no significant response either in signal intensity or localisation, maintaining the pattern described in [4]. However, ZmTCRR2 transgenic plants responded to cytokinin, displaying a very intense GUS signal, especially in the root. Significantly, the cytokinin treatment extended the GUS signal to all root tissues, especially towards the tip of the root, where the ZmTCRR2 promoter activity was not detectable in the other treatments. The hormonal treatments thus revealed an additional difference between both ZmTCRRs. While ZmTCRR1 did not respond to any hormonal stimuli, ZmTCRR2 was strongly induced by cytokinins, resembling the response obtained for canonical type-A response regulators.

### 2.3. ZmTCRR2 Interferes with Cytokinin Signalling

To further investigate the interaction between ZmTCRRs and hormone signalling, we produced Arabidopsis transgenic plants overexpressing (35S promoter) the coding sequences of ZmTCRR1 or ZmTCRR2. A striking difference in the effect of the two genes appeared in the primary transgenic lines in the glufosinate selections cultures. After 21 days of culture (Appendix A), 35S:ZmTCRR1 transgenics resemble the phenotype of WT Columbia plants concerning size, the number of leaves and flowering time. However, 35S:ZmTCRR2 plants developed tiny rosettes, always smaller than 0.5 cm, and did not flower. In rare cases (Appendix A), 35S:ZmTCRR2 plants produced small inflorescences 90 days after germination (i.e., 60 days later than the WT or 35S:ZmTCRR1 plants).

Since Type-A response regulators are usually linked to cytokinin perception pathways as negative modulators, we investigated whether the 35S:ZmTCRR2 phenotype was associated with a reduced cytokinin perception in these plants. Primary transgenic plants were selected in vitro in glufosinate-containing media and then transferred to media containing increasing amounts of cytokinin (Figure 3). In media containing no hormone, 21 day-old WT (or 35S:ZmTCRR1) plants develop regular rosettes, roots, and flower buds. In these plants, adding BA blocks flowering and promotes abnormal outgrowth of the rosettes. Cytokinin produced opposite effects on 35S:ZmTCRR2 transgenics; these plants did not flower in BA-free media, and BA induced flowering in a concentration-dependent way, with a maximum effect at 5 µM BA. Rosette leaves did not show any outgrowth in the presence of BA.

We also detected (Figure 3) that the root system of 35S:ZmTCRR2 transgenics appears to be immune to BA, while the growth of the roots of WT and 35S:ZmTCRR1 transgenics is blocked by BA in a concentration-dependent fashion. To quantify this, we monitored root growth in vertical plate assays in the absence or presence of 5 µM BA (Figure 4). Since the 35S:ZmTCRR2 transgenics were not viable, we used primary transgenic lines (56 35S:ZmTCRR1 and 37 35S:ZmTCRR2) selected for 7 days in glufosinate plates for this experiment. The presence of cytokinin (Figure 4B) severely hampered the growth of the primary root and completely blocked the formation of secondary roots in WT and 35S:ZmTCRR1 transgenics. On the other hand, 35S:ZmTCRR2 transgenics developed an almost normal primary root with secondary roots in the presence of BA, in contrast to the small and disorganised root system shown by these plants in the absence of hormone (Figure 4A). Measurements of the primary root in these plants (Figure 4C) confirm that the treatment with 5 µM BA partially rescued the root phenotype conferred by the presence of 35S:ZmTCRR2. After 3 days in the presence of BA, the primary root length of 35S:ZmTCRR2 plants is statistically indistinguishable from that of the WT and 35S:ZmTCRR1 plants grown in media with no hormone (for pairwise comparison statistics, see Appendix A). Notably, 35S:ZmTCRR1 transgenics were indistinguishable from the WT plants in these experiments.

### 2.4. ZmTCRR1 Interferes with Auxin Signalling

ZmTCRR2 behaved as a canonical type-A RR and repressed CK signalling. Plants overexpressing ZmTCRR1, on the other hand, behaved like wild type concerning CK sensitivity. Since auxin and CK are the main hormonal inputs when the ZmTCRRs are expressed, we decided to investigate the response to auxin of the Arabidopsis lines ectopically expressing ZmTCRR1.

Since 35S:ZmTCRR1 transgenics were perfectly viable, we derived two homozygous transgenic lines carrying a single insertion and showing a moderate expression level of the transgene (Appendix A), comparable to that observed for ZmTCRR1 in the maize kernel. We analysed seeds from these lines in vertical plate assays in the presence or absence of auxin (NAA). In media with no hormone (Figure 5A), the roots of the 35S:ZmTCRR1 transgenics elongated faster than the roots of the WT (Col) plants. This phenotype was reverted entirely in the presence of 125 nM NAA (Figure 5B). Primary root measurements (Figure 5C) confirm that, after 10 days in culture with NAA, 35S:ZmTCRR1 transgenics were indistinguishable from the WT plants grown in plates with no-hormone (for pairwise comparison statistics, see Appendix A). We also detected a dosage-dependent effect of the hormone treatment; the inclusion of 12.5 nM NAA in the plates partially rescued the phenotype (Appendix A).

## 3. Discussion

Two-Component Systems (TCS) pathways are instrumental in cytokinin signalling, regulating various essential processes in plant development [17,18]. However, the conserved structure of the different molecules in the system has been used along evolution to transduce other inputs in various organisms by combining module sets or minor sequence changes. Thus, TCS pathways have been shown to relay nutritional status [19], hormonal perception [20], stress [21] and a myriad of stimuli in bacteria [22]. In plants, the transcriptional control of the system relies on canonical type-B response regulators. In this context, the regulation of ZmTCRRs in the maize seed by a BETL-specific transcription factor, ZmMRP1 [3,4], was unexpected; ZmMRP1 belongs to the one-domain myb category [6] but lacks any sign of a receiver domain. This finding suggested that this transcription factor had co-opted genes that present a related binding motif in their promoters to perform functions required for proper seed development. Similar crosstalk mechanisms have been described for Wuschel and type-A response regulators in Arabidopsis apical meristem [23] or the positioning and development of protoxylem files in the Arabidopsis root [24]. Recent expression analyses using RNAseq confirm the expression overlap of ZmTCRR1 and 2 during kernel development ([14]; Figure 1A). However, a study using laser-assisted tissue dissection [15] detected ZmTCRR2 also in the conductive cells (Figure 1B), where ZmMRP1 is not present. This result opens the possibility that another Myb-related gene regulated ZmTCRR2, perhaps an unidentified canonical type-B response regulator, at least outside the endosperm transfer cells.

The analysis of Arabidopsis ZmTCRR reporter lines indicated that the promoter sequences of the genes are recognised and activated in several tissues associated with transporting structures in Arabidopsis. Muñiz et al. [4] reported the expression of ZmTCRR1 in vascular parenchyma cells of Arabidopsis plantlets, particularly in the branching point located under the apical meristem. ZmTCRR2 shows a broader expression pattern (Appendix A), with the strongest activity in perivascular regions. Type-B response regulators expressed in these tissues are probable candidates for the promoter activation. As these transcription factors participate in hormone-dependent gene activation [12], we tested the maize promoters with hormonal stimuli previously reported to be related to two-component systems [25,26]. We observed a differential response to hormones in the ZmTCRR promoters (Figure 2). None of the stimuli tested activated ZmTCRR1. ZmTCRR2, on the other hand, responded to 24 h cytokinin treatment with a marked induction, particularly in root tissues. The GUS signal increased significantly and extended to other tissues, particularly the root tip, which did not show ZmTCRR2 activity in untreated plantlets. This difference in hormonal response further separated both ZmTCRRs, suggesting that the absence of a phosphorylatable Asp residue in ZmTCRR1 might reflect a wider functional differentiation than anticipated.

In our study, ZmTCRR2 behaved as a canonical type-A molecule. It contains the DDK triad that characterizes response regulators, and we have shown that the promoter responds to cytokinin treatment in Arabidopsis reporter lines. The examination of 35S:ZmTCRR2 transgenic Arabidopsis plantlets further supported a strong inhibitory activity for ZmTCRR2 in cytokinin signalling (Figure 3, Figure 4 and Appendix A). These plants were non-viable, but the numerous transgenic lines examined showed a remarkably homogeneous phenotype, mimicking the triple mutant seedlings ahk2-1/ahk3-1/ahk4-1, which lack a functional cytokinin receptor [27]. As found here for the 35S:ZmTCRR2 transgenics, ahk2-1/ahk3-1/ahk4-1 seedlings showed very reduced size, short and thin primary roots, increased number of adventitious roots, and did not flower. A small fraction of the triple mutant plants and the 35S:ZmTCRR2 transgenics could finally bloom but were sterile (Appendix A). Using the 35S:ZmTCRR2 transgenics, we could prove that the phenotype results from cytokinin insensitivity. The treatment with BA reverted, at least partially, the lack of flowering (Figure 3) and the reduced growth of the primary root (Figure 4).

Our results reveal a remarkable repressive effect of ZmTCRR2 on the cytokinin signalling pathway in Arabidopsis. 35S:ZmTCRR2 transgenics behave as cytokinin-blind as the triple mutant ahk2-1/ahk3-1/ahk4-1. Assuming a similar CK-repressive function in maize, our results suggest that the accumulation of ZmTCRR2 in the conductive tissue of the maize endosperm reduces the activation of cytokinin signalling pathways, at least between 6 and 12 DAP when the expression of the gene peaks.

Cytokinin signalling is a critical input in seed differentiation. The kernel actively produces cytokinins, with the hormone levels peaking at very early developmental stages (0 to 8 DAP), decreasing afterward by degradation or chemical inactivation of the hormone [28,29]. In coordination with an auxin peak, this phenomenon signals the initiation of the differentiation program for internal endosperm cells, associated with endoreduplication and cell expansion [30].

Contrary to the situation observed for ZmTCRR2, ZmTCRR1 lacks some critical features that characterize type-A RR. Thus, the ZmTCRR1 promoter was not induced by cytokinin (Figure 2), and the ZmTCRR1 sequence lacks some essential residues for the phosphate signal transfer [3]. Consistently, overexpression of ZmTCRR1 in Arabidopsis did not modify cytokinin sensitivity (Figure 3 and Figure 4). However, we found that the Arabidopsis ZmTCRR1 overexpressing plants showed an auxin-insensitive phenotype, which was complemented by NAA treatment (Figure 5) in a concentration-dependent manner (Appendix A).

In this work, we have not explored the mechanism that could use ZmTCRR1 to interfere with auxin signalling; the implication of a phosphate transfer in the interaction with the auxin sensing machinery seems unlikely. However, it has been reported [31] that type-A RR might alter auxin signalling by regulating the abundance of auxin efflux transporters of the PIN class, involving a post-transcriptional regulatory mechanism.

Kernels synthesize auxin [32], and auxin signalling is a relevant input in kernel development and the differentiation of transfer cells [33]. Auxin levels are high during early developmental phases, increasing from 6 DAP over a thousand times, while the concentration decreases in the embryo [34,35]. As a prelude to nutrient storage initiation, the auxin increase is associated with active endoreduplication [36]. Significantly, the expression of auxin biosynthesis genes of the YUCCA family (ZmYUC1) has proved essential for developing the maize BETL [37,38]. If our findings in Arabidopsis were confirmed in maize, the accumulation of the ZmTCRR1 peptide in the cells positioned immediately above the transfer cells would modulate the effect of the auxin surge occurring at the transfer cells and upper endosperm.

Interactions between TCS and auxins have been previously reported [1], showing reciprocal influences in the meristem growth and organisation in maize and Arabidopsis. Abphyl1, a maize mutant deficient in the expression of ZmRR3, shows reduced ZmPIN1a expression in the shoot apical meristem, suggesting that cytokinins mediate the rapid expression of this auxin transporter through a TCS signalling mechanism [25]. In the Arabidopsis root meristem, ARR1 (a type-B RR) overexpression increases the accumulation of SHY2, an auxin-response inhibitor, via direct interaction with the SHY2 promoter. Amongst other functional consequences, the induction of the inhibitor results in the decrease in PIN transporters in the transition zone of the root, facilitating cell differentiation in this region [39].

Interestingly, in a recent study in barley [40], an endosperm-specific histidine kinase was found to be essential for transfer cell differentiation. The gene does not cluster with ethylene or cytokinin receptors, but the analysis of RNAi kernels revealed regulatory links with atypical auxin signalling elements and canonical type-B response regulators. These findings suggest the presence of yet-to-be-proved common mechanisms operating in maize and barley TC specification.

We have previously shown [3,4] that the ZmTCRR1 and 2 peptides accumulate in the inner endosperm in a transition area defined as the conducting zone or conducting tissue [10]. Cells in this region are elongated along the pedicel-silk axis and facilitate the symplastic transport of metabolites towards the central endosperm and crown during the grain filling phase. Conductive cells do not accumulate storage products until late in development, thus remaining in a juvenile stage. Remarkably, Zheng and Wang [41] suggested that ZmTCRR1 may inhibit storage compound accumulation in endosperm transmitting cells to keep the transport channel competent to support the movement of nutrients towards the crown starchy endosperm.

Further studies will be required to weigh the relative contribution of the ZmTCRR1/2 system to the regulation of the seed maturation process. Particularly, efforts are underway to produce mutant kernels for ZmTCRR1, 2, or both using CRISPR/CAS aided gene editing.

However, the results and materials presented here will be essential for characterising the maize mutants. Notably, the Arabidopsis lines would facilitate the rapid verification of any mechanistic model suggested by the maize phenotypes by allowing physiological experiments that would not be very feasible in immature maize kernels.

### Conclusions

Type-A response regulators are negative modulators in two-component hormone-signaling pathways. In this study, using the Arabidopsis heterologous system, we have found that ZmTCRR-2 is a strong negative modulator of the cytokinin signalling, while ZmTCRR-1 has no measurable effect in this pathway. Since the cytokinin signalling pathway is highly conserved in plants, it is reasonable to expect a similar physiological effect for these molecules in maize. Therefore, this study provides the grounds for elaborate models of the function of these molecules in endosperm development. However, since the presence of endogenous molecules that can alter the interaction of the ZmTCRRs with the pathway cannot be excluded, the confirmation of the Arabidopsis data would require the examination of maize plants manipulated for the expression of the ZmTCRRs.

## 4. Materials and Methods

### 4.1. RNA/DNA Extraction

DNA was extracted using a cetyl-trimethylammonium-bromide (CTAB) based buffer (Weigel and Glazebrook, 2002).

RNA was extracted using the Maxwell 16 LEV Plant RNA Kit (Promega, Madison, WI, USA) following the manufacturer’s instructions. All samples were checked for quality on a 1.5% denaturing agarose gel.

### 4.2. Constructs Design and Preparation

ZmTCRR1 and 2 promoter constructs are described in [4]. For ubiquitous expression of ZmTCRR1 and 2, the coding sequences were amplified from seed cDNA (maize variety A188), cloned into pEntry-D/TOPO vectors (Invitrogen, Thermo Fisher Scientific Inc. Waltham, MA, USA) and then transferred to pEarley100 [42] using Gateway technology.

The final constructs were then transferred to Agrobacterium tumefaciens C58C1 for Arabidopsis transformation.

### 4.3. PCR and QRT-PCR Assays

All primers used in this study are shown in Appendix A.

For real-time PCR, we prepared cDNA from 500 ng RNA in 10 µL reactions (PrimeScript RT reagent Kit, Takara, Kusatsu, Japan). Approximately 10 ng cDNA were used in 10 µL real-time PCR reactions with SYBR Premix Ex Taq (Takara Bio Europe S.A.S, St Germain en Laye, France). Reactions were run and analysed in a Rotorgene-Q machine (Qiagen, Düsseldorf, Germany).

### 4.4. Arabidopsis and Maize Transformation

Transgenic Arabidopsis lines were generated following the floral dip protocol [43].

### 4.5. Arabidopsis Culture

Unless otherwise stated, all cultures were performed under a long-day light regime. Surface-sterilised seeds were plated on MS 1×, 0.8% agar, 1% sucrose, and stratified for 2 days at 4 °C, to be later moved to the growth chamber. All additional chemicals were added at the time of plate preparation. Hormones (ACC, AVG, BA, gibberellic acid, and abscisic acid) were purchased from Sigma (Sigma, Merck group, Burlington, MA USA).

### 4.6. Vertical Plate Assays

For the experiments with primary transgenics, surface-sterilised seeds were plated on MS 1×, 0.8% agar, 1% sucrose, 30 µg/mL ammonium glufosinate, and stratified for 2 days at 4 °C, to be later moved to the growth chamber. After 7 days, transgenic plants were identified as they started producing true leaves and developed roots in the culture media. These plants were transferred to squared plates containing media with no herbicide and the indicated amount of BA or mock.

For the experiments involving homozygous lines overexpressing ZmTCRR1 or Columbia wild-type plants, seeds were plated in the media described above with no Phosphinothricin. After 1 week in culture, plantlets were transferred to squared plates containing media with the indicated amount of NAA or mock.

Squared plates were cultured in a vertical position, and pictures were taken every two days. Root growth was quantified using the SmartRoot software [44].

### 4.7. Statistical Analyses

Statistical analyses were performed using the package Statgraphics Centurion V19. Data from 15–30 plants per genotype and treatment were used for the analyses. Levene’s test revealed inequality of variances among sample groups, and comparisons between groups were consequently made using non-parametric approaches. Bonferroni’s test was used for the comparison of sample pairs. Results from each statistical analysis are compiled in Appendix A.

### 4.8. Histology

GUS activity detection was performed by immersing the plantlets in a buffer containing potassium ferro- and ferricyanide (0.5 mM each), 50 mM sodium phosphate, 10 mM EDTA, and 0.1% Triton X-100, 10% methanol, and 1 mg/mL X-GLUC (Duchefa). After overnight staining, Arabidopsis plantlets or tissue samples were fixed and embedded in LR White resin (Sigma) following a protocol from Dr. Nicholas Harris (Dept. of Biological Sciences, U. Durham, ftp://ftp.arabidopsis.org/home/tair/Protocols/compleat_guide/2_fix_and_embed.pdf, accessed on 26 July 2022). Resin pieces were sectioned at 2 µm thickness and counterstained with 2% aqueous fuchsine. Light and fluorescence microphotographs were taken using a B-600TiFL microscope (Optika, Ponteranica (BG), Italy), and whole plantlets were photographed under a Stemi 2000-C stereomicroscope (Zeiss, Oberkochen, Germany).

## Figures and Tables

**Figure 1 plants-11-01992-f001:**
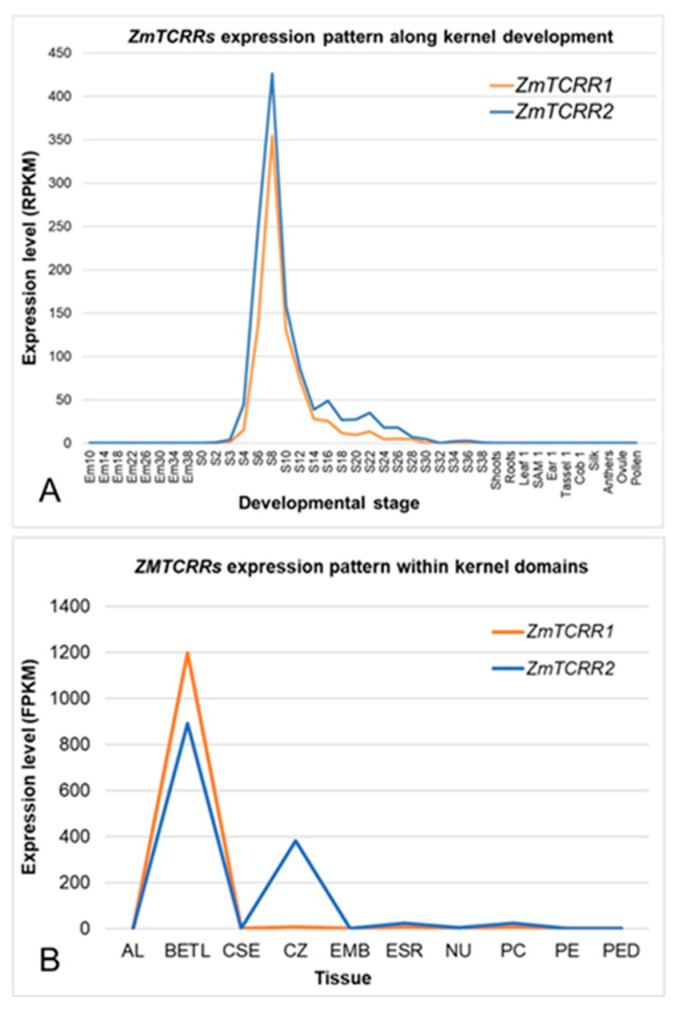
RNAseq expression analyses of ZmTCRR1 and ZmTCRR2. (**A**) The developmental expression pattern of these genes was analysed using the RNAseq data from the Chen et al. study [14]. Em10-38, embryo at the indicated DAP developmental stage; S0-38, kernel at the indicated DAP developmental stage. The last 11 samples represent other non-kernel plant tissues. SAM, shoot apical meristem. (**B**) The expression pattern of the genes was extracted from the 8 DAP RNAseq anatomical study by Zhan et al. [15]. Al, aleurone; BETL, basal endosperm transfer cell layer; CSE, central starchy endosperm; CZ, conductive zone; EMB, embryo; ESR, embryo surrounding region; Nu, nucellus; PC, placenta-chalaza; PE, pericarp; PED, pedicel.

**Figure 2 plants-11-01992-f002:**
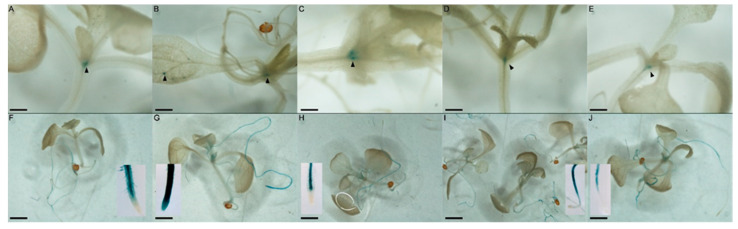
Cytokinins induce the ZmTCRR2promoter in Arabidopsis. Upper panels (**A**–**E**) show results for the ZmTCRR1 promoter, and lower panels (**F**–**J**) display ZmTCRR2 promoter plants. seven-day-old plantlets were cultured for 24 h on 1 µM ACC (**A**,**F**), BA (**B**,**G**), IAA (**C**,**H**), gibberellic acid (**D**,**I**) or MS control plates (**E**,**J**), and then stained for GUS activity. Distained plants were photographed to assess promoter activation qualitatively. For the ZmTCRR1 promoter, none of the treatments caused any observable change in staining, either in intensity or location. On the other hand, ZmTCRR2 responded to cytokinins with an increase in the intensity of the signal in the root, while other hormonal activators did not cause significant changes under these culture conditions. Note that the root tip remains unstained in all treatments but BA. Insets in (**F**–**J**) show the higher magnification images of the primary root tip. Scale bars represent 0.5 mm in (**A**–**E**) and 2 mm in (**F**–**J**).

**Figure 3 plants-11-01992-f003:**
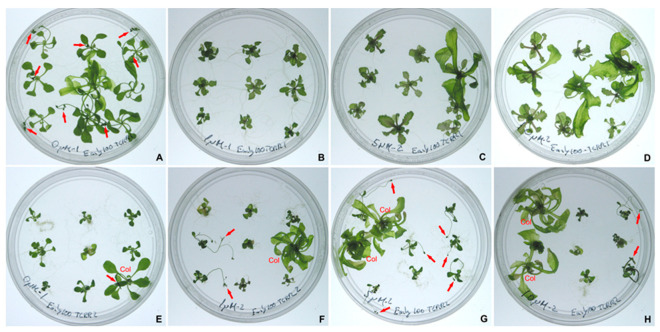
Cytokinin responses in ZmTCRR1 or 2 overexpressing plants. Seven days old primary transgenic plants overexpressing (35Spromoter) ZmTCRR1 (**A**–**D**) or ZmTCRR2 (**E**–**H**) were cultured in MS media containing 0 µM (**A**,**E**), 1 µM (**B**,**F**), 5 µM (**C**,**G**) or (**D**,**H**) 10 µM BA for 16 days. Some plants were found non-transgenic by DNA genotyping and thus labelled as Col (Columbia) in the figure. Arrows point to flower buds. The plates are 90 mm diameter Petri dishes.

**Figure 4 plants-11-01992-f004:**
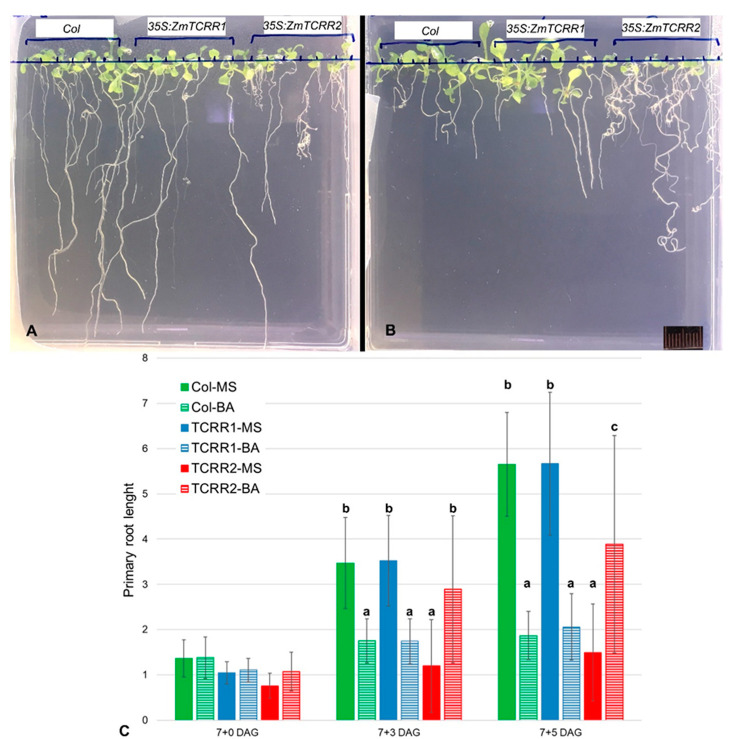
ZmTCRR2, but no ZmTCRR1, interferes with cytokinin signalling. Seven-day-old primary transgenic plants were extracted from the herbicide selection plates and cultured vertically with MS media containing no hormone or 5 µM BA. Upper panel, a representative set of plates, 7 days after culture. (**A**), no hormone; (**B**), 5 µM BA. The rule in (**B**) is 1 cm. (**C**), average primary root length (in cm) of each genotype in each culture media, after 0, 3, or 5 days in the vertical plates. The error bars represent the standard deviation. The graph shows a strong interaction between genotypes and treatments. After 3 days of culture, ZmTCRR2 plants in the presence of 5 µM BA are in a statistically homogeneous group with WT (Col) and ZmTCRR1 plants in the absence of hormone; this group is statistically distinct (*p* < 0.05) from that formed by WT and ZmTCRR1 plants in the presence of the hormone. Letters on the columns identify statistically homogeneous classes according to pairwise comparisons (Bonferroni test, 95% confidence).

**Figure 5 plants-11-01992-f005:**
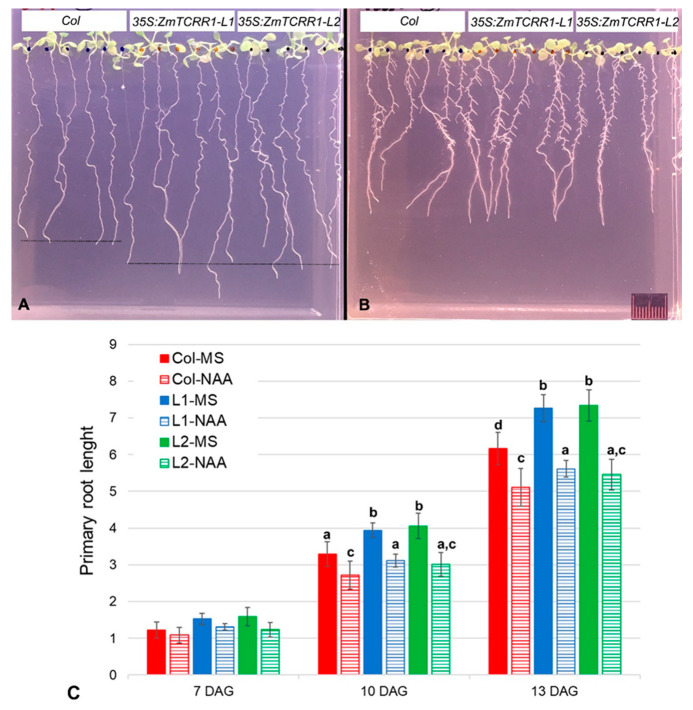
ZmTCRR1 interferes with auxin signalling. Seeds from wild type (Col) and two independent homozygous transgenic lines (L1, L2) overexpressing ZmTCRR1 were cultured in vertical plates with MS media containing no hormone or 125 nM NAA. The upper panel shows a representative set of plates after 13 days in culture. (**A**), no hormone; (**B**), 125 nM NAA. Dot lines in (**A**) highlight the difference in growth between Col and the transgenic lines. The rule in (**B**) is 1 cm. (**C**), average primary root length (in cm) of each genotype in each culture media after 7, 10, and 13 days after germination (DAG). The error bars represent the standard deviation. The transgenic lines show nearly identical behaviour and a consistently faster primary root growth than the WT plants in the absence of hormone. However, after 10 days of culture, ZmTCRR1 plants in the presence of 125 nM NAA are in a statistically homogeneous group with WT (Col) plants in the absence of hormone, and this group is statistically distinct (*p* < 0.05) from that formed by the ZmTCRR1 plants in the absence of hormone. According to pairwise comparisons, letters on the columns identify statistically homogeneous classes (Bonferroni test, 95% confidence).

## Data Availability

Not applicable.

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
