# Peer review of "Distinct Hormone Signalling-Modulation Activities Characterize Two Maize Endosperm-Specific Type-A Response Regulators"

_plants, 2022, doi:10.3390/plants11151992_

Round 1

Reviewer 1 Report

The manuscript described gene expression analysis of ZmTCRR1 and 2 in maize kernel tissues, and results from overexpression lines in Arbidopsis. To me it is not clear how to draw a general conclusion of the work-of-action of these transcription factors in maize kernels from results that were obtained in a completely different tissue and species. Even though the authors stated the reason why not performing the same results in transgenic maize, but to me that still does not explain how to translate a function that should appear in maize kernels with results obtained mainly in Arabidopsis roots. Thus, I don´t think that general conclusions drawn in the manuscript are accurate (incl. for example line  19-21 or line 310-314). the logical or biological basis that would justify transfering results from Arabidopsis roots to a potential work of action mode in maize endosperm is not described. I would therefore suggest to take out any conclusions that suggest a potential mode of action or function of both transcription factors in maize endosperm. 

Author Response

We thank the reviewer for their comments.

We completely agree with the reviewer that no conclusions regarding the function of these molecules in endosperm cells function or differentiation can be drawn from the data obtained in Arabidopsis. The work was not intended for that purpose, and we have revised the ms to eliminate any statement suggesting such conclusions.

This work aimed to test the molecular function of the ZmTCRR proteins in the context of the highly conserved signal transduction mechanisms such as the CK pathway. We believe it is safe to predict from the Arabidopsis data that, in maize, ZmTCRR-2 will interfere with CK signalling while ZmTCRR-1 will not. Indeed, that should be the situation if we consider the conservation of the remaining components of the pathway between Arabidopsis and maize. However, we agree that the presence of a yet undescribed element in maize that would interact with the type-A response regulators unexpectedly cannot be excluded. We have tried to acknowledge all these possibilities in section 3.1, “Conclusions”, in the revised ms.

Reviewer 2 Report

The paper is generally well written and structured. ZmTCRR1 and ZmTCRR2 are two endosperm-specific type-A response regulators, their promoters respond differently to hormonal signals. ZmTCRR1 interferes with auxin signaling, while ZmTCRR2 participates in the cytokinin pathway. The mechanisms of crosstalk between cytokinin and auxin are beyond question, however, the specific function of these two hormones in the endosperm is still not clear. This paper supports a regulatory mechanism that coordinates cytokinin and auxin perception in the inner endosperm cells.

In my opinion, the paper still has one shortcoming in regards to some data analysis and text. The author has analyzed the expression level of ZmTCRRs using the conventional technique (RT-PCR) before, so we want to know if the result is consistent with the RNAseq expression analysis of ZmTCRR1 and ZmTCRR2?

Author Response

We thank the reviewer for their comments.

Yes, we found the same results using RT-PCR compared with the RNAseq. We clarify this in lane #108 in the revised ms.